# Cortical and autonomic responses during staged Taoist meditation: Two distinct meditation strategies

**Maria Volodina[1], Nikolai Smetanin[1], Mikhail Lebedev[1], Alexei Ossadtchi[1,2]***

**1** Center for Bioelectric Interfaces, HSE University, Moscow, Russia, **2** Artificial intelligence Research Institute, Moscow, Russia

* aossadtchi@hse.ru

**Data Availability Statement:** All relevant data are within the paper and its Supporting information files.

## Abstract

Meditation is a consciousness state associated with specific physiological and neural correlates. Numerous investigations of these correlates reported controversial results which prevented a consistent depiction of the underlying neurophysiological processes. Here we investigated the dynamics of multiple neurophysiological indicators during a staged meditation session. We measured the physiological changes at rest and during the guided Taoist meditation in experienced meditators and naive subjects. We recorded EEG, respiration, galvanic skin response, and photoplethysmography. All subjects followed the same instructions split into 16 stages. In the experienced meditators group we identified two subgroups with different physiological markers dynamics. One subgroup showed several signs of general relaxation evident from the changes in heart rate variability, respiratory rate, and EEG rhythmic activity. The other subgroup exhibited mind concentration patterns primarily noticeable in the EEG recordings while no autonomic responses occurred. The duration and type of previous meditation experience or any baseline indicators we measured did not explain the segregation of the meditators into these two groups. These results suggest that two distinct meditation strategies could be used by experienced meditators, which partly explains the inconsistent results reported in the earlier studies evaluating meditation effects. Our findings are also relevant to the development of the high-end biofeedback systems.

## Introduction

Meditation is an ancient tradition that dates back thousands of years. It can be described as a complex process aimed at self-regulating the body and mind [1]. The popularity of meditation in the Western world has risen significantly over the past decades [2]. This can be explained not just by fashion but also by a growing body of evidence on the meditation benefits. To list a few, meditation has been shown to reduce depression symptoms, anxiety, pain [3, 4], decrease stress level and improve attention [5], promote emotional regulation, prosocial behavior and subjective well-being level [6], reduce cardiovascular risk [7], decrease the glucocorticoids level and expression of pro-inflammatory factors in the blood [8, 9]. Meditation-based

**Funding:** This work is supported by the Center for Bioelectric Interfaces NRU HSE, RF Government grant, AG. No. 075-15-2021-624. The funders had no role in study design, data collection and analysis, decision to publish, or preparation of the manuscript.

**Competing interests:** The authors have declared that no competing interests exist.

techniques are already part of academic programs in many American and European schools [10, 11], as well as medical practice [12–14].

However, besides the proven beneficial effects, meditation can also cause an unpleasant experience or even adverse effects [15] like uncomfortable kinaesthetic sensations, mild dissociation, feelings of guilt, destructive behavior, suicidal feelings [16], and transient psychosis [17]. There is also an opinion that meditation could predispose to seizures [18].

Most of these adverse effects are transient and disappear over time. However, some of them, like suicidal intentions, can be really dangerous; in these cases meditative practice should be interrupted or changed. This is one of the reasons why traditionally one learns to meditate under supervision of an experienced person. In addition to giving recommendations about the process (like a proper body position etc), such a person ensures that meditation effects are proper and not dangerous. Today, with the rise of popularity of meditation, more people start practicing it on their own without any supervision. This situation increases the need for developing a clear and objective set of criteria that would hallmark a proper meditation process and distinguish it from malpractice. Once developed and thoroughly tested, these criteria may be embodied into high-end devices providing automatic biofeedback during meditation sessions.

Currently, existing biofeedback systems for meditation are based on various physiological indices. Some of them use autonomous nervous system (ANS) markers, others employ markers of central nervous system (CNS) activity, and some a combination of both. Among biofeedback systems reported previously, there are those that use heart rate data obtained by photoplethysmography [19], respiratory rate [20], breathing awareness (subjects were asked to put marks on a certain phase of breathing, and the marks affixed by the user were compared with objective data on the breathing phase) [21], and galvanic skin response [22]. Neurofeedback systems include neurofeedback derived from fMRI of the posterior cingulate cortex [23], EEG power in the gamma band during transcendental meditation [24], and mindfulness meditation [25], and EEG power in various bands recorded simultaneously with electrocardiography (ECG) data [26].

At present, there is also a wide range of consumer-level bio- and neurofeedback devices available over the counter. Although there is some evidence of the benefits of meditation assisted with such devices [27], most of them do not have a proven advantage over the non-feedback-assisted meditation. The obvious problem in the development of such assistive devices is the lack of consensus regarding the detection of different meditation states and the selection of physiological markers to be used to form the feedback signal. Despite decades of intensive research, physiological effects of meditation are still unclear. This can be easily explained by a wide range of meditation types explored in different studies and by a very significant interindividual variability between subjects [28, 29]. This study considers only the mindfulness meditation.

As outlined above, the research data about physiological changes during meditation are inconsistent and partly depend on the meditation type. Next, we briefly summarize previous findings regarding the effects of mindfulness meditation. Lomas et. al. [30] concluded that the increased power in the alpha and theta bands was a hallmark of most mindfulness meditation studies. It is noteworthy that in the studies with meditation-naive subjects [31] no differences were found between mindfulness and a resting state. Similar observations were also made in the studies with the experienced ones [32, 33]. Reduction of alpha power during mindfulness was also observed [34]. Simultaneous increase in the power of alpha and theta rhythms associated with a meditative state found in most studies could indicate the state of "relaxed alertness" [35]. The described effects of meditation on the power of beta, delta, and gamma rhythms differ across the mindfulness meditation studies and do not represent a coherent picture [30].

The results of different studies exploring the changes in the activity of the autonomic nervous system during a number of mindfulness meditation types also vary a lot. However, most of them consistently report the evidence for a shift of sympathetic/parasympathetic system balance towards parasympathetic activation. The main findings include the increased heart rate variability (HRV) [36–38] decreased systolic blood pressure [8, 39], decreased [8, 40] or increased heart rate [41], decreased respiratory rate [42], and increased skin resistance [43] during meditation.

All together, these data, as well as the subjective personal reports of experienced meditators and Traditional Buddhist formulations, support the point of view that the subjects experience "relaxed alertness" state during meditation and should avoid both excessive hyperarousal (restlessness) and excessive hypoarousal (drowsiness, sleep) [35].

Based on this analysis of the previous literature, we conclude that there is a lack of studies exploring gradual changes of physiological indicators during meditation as well as association of specific indicators with meditation depth. Also, while a few studies performed comparative analysis of effects of different kinds of meditation [34, 44] none of them discussed the possibility of existence of different strategies for performing the same type of meditation. Such data would contribute both to the knowledge of fundamental basics of physiology of meditation and to the development of high-quality devices for biofeedback assisted meditation.

To fill this gap and to gain knowledge about the dynamics of changes in the activity of the nervous system, we asked our subjects to perform a stage-by-stage guided Taoist meditation that could be classified as mindfulness meditation. It consisted of several consecutive stages including relaxation, body scan, stopping of internal dialogue, visualization, and instructed breathing. The participants in the experienced meditators group regularly practised meditation for two years or more. Subjects without any meditation experience were included in the novices group and followed the same instructions as the experienced meditators.

When selecting subjects for the experienced meditators group, we did not limit the style of regular meditation they practiced. All of our experienced participants appeared to be experts in different types of meditation. None of them, except for one person, had a regular experience in Taoist meditation, which was used in the experiment. So, both experienced meditators and novices read the text before the experiment to familiarise themselves with it, but did not know the text by heart and just followed the instructions. With such a design we are able to tackle the question of whether people who regularly practice meditation develop the ability to consciously control their physiological state in any situation beyond the particular meditation practice. Our design also removes the potential systematic bias of familiarity with the text in the experienced meditators group.

Our goal was to explore the dynamics of CNS and ANS activity during a step by step guided meditation and compare it between the experienced meditators and the subjects without a meditation experience. We expected to see gradual changes of physiological markers indicative of a simultaneous increase in mind concentration (EEG) and peripheral relaxation (ANS markers) from the beginning to the middle of a meditation session and a gradual return to the baseline during the second part of the session. This hypothesis was partly based on the subjective reports from the experienced meditators and meditation teachers about gradual entering into meditative state and subsequent gradual emerging. Standard recommendation by meditation teachers is to aim at a similar duration of entering and emerging. Also, the U-shape profile hypothesis appears to be consistent with the data by DeLosAngeles et al., 2016 who obtained such characteristic profiles in theta, alpha and beta power in their study [45]. We hypothesized that such changes would be more pronounced in the experienced meditators than in the novices.

## Experimental design, materials, and methods

### Participants

Overall, 28 subjects took part in the experiment (13 –experienced meditators, 15 –novices, 2 more subjects from the experienced meditator group were excluded because of technical problems during recording). The age of the experienced meditators ranged from 27 to 52 years (mean age = 37.3 ± 8.9) and the age of novices ranged from 25 to 55 years (36.0 ± 7.3). The number of male/female participants was 8/5 and 6/9 in the experienced and novice groups respectively. The inclusion criterion for the experienced meditators group was regular (several times a week) meditative practice for 2 years or more. Novices should not have any experience in meditation.

The experienced meditators were not restricted to practicing a specific type of meditation and had experience with various kinds of meditation. Most of the meditators (11 out of 13) reported regular practice of more than one type of meditation.

The experiment was conducted in accordance with the declaration of Helsinki. Participation in the study was voluntary. All participants provided written consent approved by The Higher School of Economics Committee on Inter-University Surveys and Ethical Assessment of Empirical Research in accordance with the Declaration of Helsinki. All experimental protocols were approved by The Higher School of Economics Committee on Inter-University Surveys and Ethical Assessment of Empirical Research in accordance with the Declaration of Helsinki.

### Experimental protocol

The experimental protocol comprised the following stages: 1) Participants read and signed the informed consent; 2) Participants answered the questionnaire, included questions about: age, gender, BMI, leading hand, meditation experience, habitual physical activity, habitual sleep duration, mood, anxiety, sleepiness, wellbeing level; 3) Participants read meditation instructions and the supplement materials with recommendations; 4) The experimenter placed the EEG cap, respiration (RESP), Photoplethismography (PPG) and Galvanic Skin Response (GSR) sensors in their appropriate places on the subject's body, described in detail later in this article; 5) The experimenter measured blood pressure and heart rate using a tonometer; 6) Participants then performed meditation guided by audio instruction, delivered through the ear-plugged headphones. The first instruction prescribed keeping still with eyes-closed for 2 minutes and was used to collect baseline resting state data; 7) Right after the full meditation session, the experimenter measured blood pressure and the heart rate again; 8) The participant answered the second questionnaire.

### Meditation guideline

The meditation guideline consisted of 16 stages. The first one was pre-meditation resting state. The guideline was provided by the experienced meditation teachers and adapted to meet the experimental needs. The end of each audio instruction marked the beginning of a new meditation stage that required executing the instruction within a specific time window. Mostly, each stage lasted for two minutes. Every minute within a stage, we presented the reminders. Such a design allowed us to exclude, from the subsequent analysis, the periods when the instruction was uttered. The instructions used and the duration of time windows allocated for executing the instruction (stage duration) are listed below:

Premeditation resting state (2 min)—"Take a comfortable position. Close your eyes."

*stage 1* (2 min)—"starting the meditation. Sit up straight. The spine is straightened. The vertex reaches for the sky, the sacrum pulls down. Relax the muscles of the face. Relax the forehead, temples, jaws, chin, muscles of the neck, clavicle. Shoulders, elbows flow down. Relax the ribs. With each breath make them softer and softer. Relax the stomach, lower back. Breathing becomes quieter, deeper."

*stage 2* (2 min)—"Bring your attention to the sacrum. From the sacrum, slowly go up, increasing the distance between the vertebrae. Glide up: 5th lumbar vertebra, 4th, 3rd, 2nd, 1st. Stretch your thoracic spine vertebra by vertebra: the 12th, 11th, 10th, 9th, 8th, 7th, 6th, 5th, 4th, 3rd, 2nd, 1st. We melt the muscles along the spine, the tendons that hold it. Glide, grow up. Shoulders, elbows relaxed, flow down. The ribs are soft. We extend the neck, also vertebra by vertebra. Slowly but persistently. Slide with your attention, grow up. 7th cervical vertebra, 6th, 5th, 4th, 3rd, 2nd, 1st. Stay in this state."

*stage 3* (2 min)—"Imagine you are hanging, hooked by the top of your head. Pull up. Pull with all your force. Even stronger. Stronger. You are hanging like an empty bell. The spine is stretched. The sacrum stretches down, stretching the spine with its weight. The spine is stretched between the vertex and the sacrum. Stay in this state."

*stage 4* (2 min)—"The tongue touches the palate. Energy flows to form a platform at the level of the third eye. The platform is becoming more powerful, its gravitational force is growing. Effortlessly, put your mind on the platform. It can be effortlessly fixated on the platform. Stay in this state."

*stage 5* (2 min)—"Relax your mind. Let it go out of your body. Let it fill the space of the room."

*stage 6* (1 min)—"Put your attention to the borders of your body."

*stage 7* (2 min)—"Look inside yourself. Let your mind observe internal emptiness. Relax your mind in the internal emptiness. Stay in this state."

*stage 8* (1 min)—"Repeat after me: "The sky above me is open and endless." See the sky. Feel its infiniteness."

*stage 9* (1 min)—"Repeat to yourself: "The earth is solid, powerful." Feel the hardness and power of the earth."

*stage 10* (4 min)—"Repeat after me: "I am sitting between heaven and earth. Like a pillar supporting heaven, filling the space between heaven and earth. Powerful." Feel yourself powerful, sitting on the earth, supporting the heavens. Stay in this state."

*stage 11* (2 min)—"Fill your body even more. The qi, shen becomes infinite."

*stage 12* (1 min)—"Feel the borders of your body."

*stage 13* (1 min)—"Put your attention to the point between the eyebrows, to the tip of the nose, to the center of the chest, to the solar plexus, to the lower abdomen. From the surface of the abdomen, turn your attention inside. Breathe naturally."

*stage 14* (2 min)—"Keep your attention in your abdomen. Retract your abdomen while inhaling, expand while exhaling."

*stage 15* (1 min) (postmeditation resting state)—"Breathe normally. Expand your abdomen while inhaling, retract while exhaling."

When you are ready, you can open your eyes and finish the meditation.
For the subsequent statistical analysis we merged these intervals into the following 6 stages:

i) Resting state: Premeditation resting state

ii) Stage 1: Merged stages 1–4 (relaxation, body scan, taking position)

iii) Stage 2: Merged stages 5–7 (stopping internal dialogue)

iv) Stage 3: Merged stages 8–10 (visualization)

v) Stage 4: Merged stages 11–14 (coming back, focus on breathing and body)

vi) Stage 5: Postmeditation resting stage

As we mentioned above all the participants read the text of meditation before the experiment and had their questions answered. The experimenter also explained the most frequent questions, e.g. where sacrum is and what is "qi" and "shen". Nonetheless, the fact that most novices had no deep understanding of philosophical concepts "qi" and "shen" is a potential limitation of this study. Another important point concerns the instruction to stretch vertebrae one by one. Here we did not expect that someone could feel a particular vertebra. This instruction was designed to help the subject gradually move attention along the spine, stretching it up. Using sequential naming of spine segments helped us to control the pace at which the subjects move their attention up the spine.

## Materials and equipment

EEG data was acquired using a 32 channel wireless EEG system (SmartBCI, Mitsar, Russia) at 250 Hz sampling rate. Digital averaged ear signal was used as a reference for all EEG data channels. We used a Python script to launch simultaneously audio instruction and EEG data acquisition. Thus, EEG data is accurately synchronized with audio instruction. The audio was provided through standard ear-plugged headphones.

GSR, PPG and RESP data was acquired using the PolyRec system (Medical computer system, Russia). Polyrec recording was manually launched by the experimenter simultaneously with the python script.

Galvanic skin response was measured using the GSR sensors (Medical computer system, Russia). Sensors were placed on the first (proximal) phalanges of the ring and index fingers of the left hand. The following filters were applied to the data during recording: 4th order 10 Hz low-pass Butterworth filter, notch filter 50 Hz. In one subject from the Novices group, filter settings for GSR were set incorrectly, so the GSR data for this subject was not included in analysis.

The Photoplethysmography sensor (Medical computer system, Russia) was placed on the index finger of the subject's right hand. The following filters were applied to the data: 4th order 0.5 Hz high-pass filter, 4th order 10 Hz low-pass Butterworth filter, notch filter 50 Hz.

Respiration was measured by a thermometric sensor of nasal respiration that was placed under the nose, according to the instruction of the provider (Medical computer system, Russia). The data were then band-pass filtered within 0.05 Hz—10 Hz using a 4th order Butterworth filter and also a 50 Hz notch filter was applied to further suppress powerline artifact.

In addition, blood pressure and pulse rate were measured, before and after meditation sessions using a medical tonometer (A&D, Japan).

## Data pre-processing

EEG data was filtered by using a bandpass filter with a lower cut-off of 1Hz and a higher cut-off at 40Hz, a 50 Hz notch filter was also applied. Independent component analysis was performed and eye movement and muscular components were rejected. We removed

components based on several main criteria: 1) high mutual info coefficient with Fp1 and Fp2 channels and absence of peak in the alpha band; 2) high level of high frequency activity. These operations were done using NFBLab software [46].

## Data processing

All the following data processing procedures were performed using MNE-python.

**EEG data processing.** EEG power spectra were computed using the Welch method, using a window size of 1s with no overlap for each meditation stage separately. Next, we averaged power spectral density profiles within the following five bands: Delta: 1–4 Hz, Theta: 4–8 Hz, Alpha: 8–14 Hz, Beta: 14–25 Hz, and Gamma: 25–40 Hz. Additionally, using the obtained stage specific mean band power values, we also computed alpha/theta and alpha/beta EEG power ratios.

To decrease the number of multiple comparisons in the subsequent statistical analysis, we used the same reduction scheme as [47] (Fig 1) and merged 30 EEG channels into 13 topographic brain regions.

**PPG data processing.** We detected heartbeats using a proprietary peak detection algorithm and calculated the heart rate (HR) and the heart rate variability (HRV) indices. We used pulse to pulse interval (RR) time series to calculate HRV indices. HR was calculated using a 20 sec long moving window with 5 sec step. For HRV indices calculation, a sliding window, with length 60 sec and step 30 sec was used. We then averaged the obtained values within each stage.

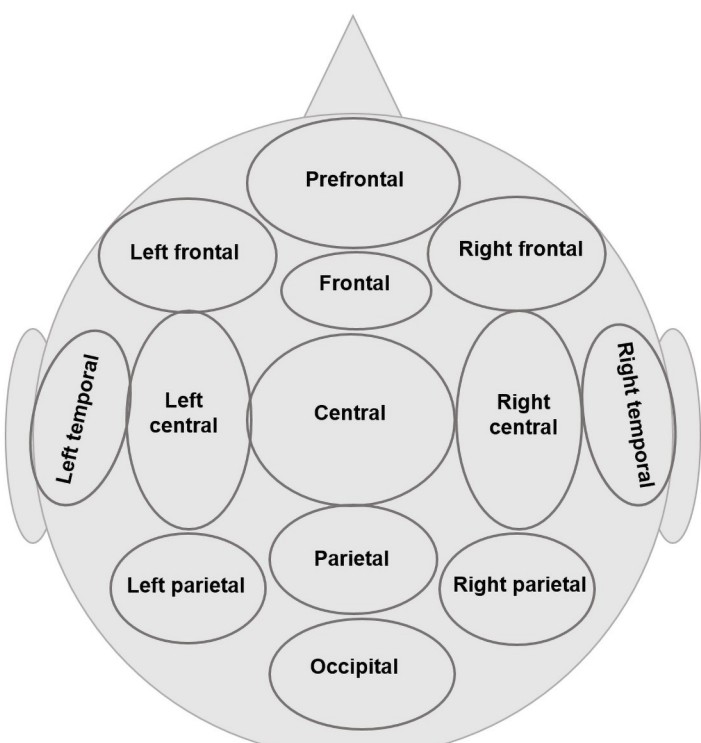

**Fig 1. Reduction scheme into 13 major topographic brain regions.** Prefrontal—Fp1, Fp2; Left frontal—F7, F3; Right frontal—F4, F8, Frontal–Fz; Central—Cz, Cp1, Cp2, Fc1, Fc2; Parietal–Pz, Left temporal—T7; Right temporal—T8; Left central—C3, Fc5, Cp5; Right central—C4, Fc6, Cp6; Left parietal—P7, P3, Po3; Right parietal—P4, P8, Po4; Occipital—O1, O2, Oz.

HRV analysis is one of the commonly used methods to appraise the activity of the sympathetic and parasympathetic nervous system [48]. HRV based indicators include time domain and frequency domain indices.

We first calculated the standard time-domain HRV indices [48]. Briefly, indices are defined as follows: 1) dRR = max(RR) − min(RR); 2) RRNN = $\frac{1}{N} sum(RR\_i)$); 3)

$$SDNN = \sqrt{\frac{1}{N-1}\sum_{N=1}^{N}\left(RR_n - \underline{RR}\right)^2};$$ 4) coefficient of variation(CV) = $\frac{SDNN*100\%}{RRNN}$; 5) ME = median

(RR); 6) amplitude of median (AME) = $\frac{(number\ of\ RR\ in\ range\ ME\pm median(RR-median))*100\%}{number\ of\ RR\ intervals}$; 7)

$$RMSSD = \sqrt{\frac{1}{N-1}\sum_{N=1}^{N-1}\left(RR_{n+1} - RR_n\right)^2};$$ 8) stress index (SI) = $\frac{AME*100\%}{2*ME*dRR}$. In the above N stands for

the number of RR intervals.

Then, we calculated frequency domain HRV indices using FFT of the pulse interval sequence [49]. The area under the curve of the spectral peaks within the frequency range of 0.04–0.15, and 0.15–0.40 Hz were defined as the low-frequency power (LF), and high-frequency power (HF). Normalized LF (LFnu) and HF (HFnu) were calculated as LF/(LF+HF) and HF/(LF+HF) respectively. Also LF/HF ratio and peak values of the spectrum in the low and high frequency bands were calculated.

LFnu can be considered as an index of combined sympathetic and vagal modulation [50] and an index of baroreflex [51], and HFnu corresponds to an index of vagal modulation. The LF/HF is sometimes considered as the index of sympathovagal balance [49].

**GSR data processing.** Galvanic skin response depends on the transient change in the skin conductivity associated with the sweat gland activity and elicited by any stimulus that evokes an arousal or orienting response. GSR can be used as a sign of arousal and activation of a sympathetic system [52]. Order 4 zero-phase high-pass Butterworth filter, with 0.05 Hz cut-off frequency, was applied to the data to remove the slow trend. We then measured the average level of GSR, number of spontaneous reactions and area under the curve. Spontaneous reactions were defined as signal fluctuations with amplitude greater than one standard deviation of the signal computed over the entire recording in each particular subject. The area under the GSR signal curve was used as a cumulative index of the count and the amplitude of spontaneous reactions. A moving window with length 60 sec and step 5 sec was used.

**RESP data processing.** A proprietary peak detection algorithm was applied to the output of our respiration sensor. The respiration rate was calculated using a moving window (20 sec length, 5 sec step). All epochs within a stage were averaged. Respiration amplitude was calculated as the average difference between the upper and lower envelopes of the signal.

## Statistical analysis

General outline of statistical analysis is presented in Fig 2.

First, group-related differences between the experienced and novice groups for pre- and post-meditation values and questionnaire scores were examined, using Student t-test (for normally distributed data) or Mann-Whitney U test (for non-normally distributed data). The complete list of analysed indicators of CNS and ANS activity can be found above, in the Data processing sections.

Next, all indicators data were normalized to individual baselines by dividing by premeditation resting state value. In the case of EEG, each rhythm's envelope and band power indices were normalized separately. We used the adaptive Neyman test (AN-test, [53]) to perform intergroup comparison of these normalized dynamics for each indicator. Unlike a full blown ANOVA test, the AN test considers temporal smoothness of curve points and exploits it to

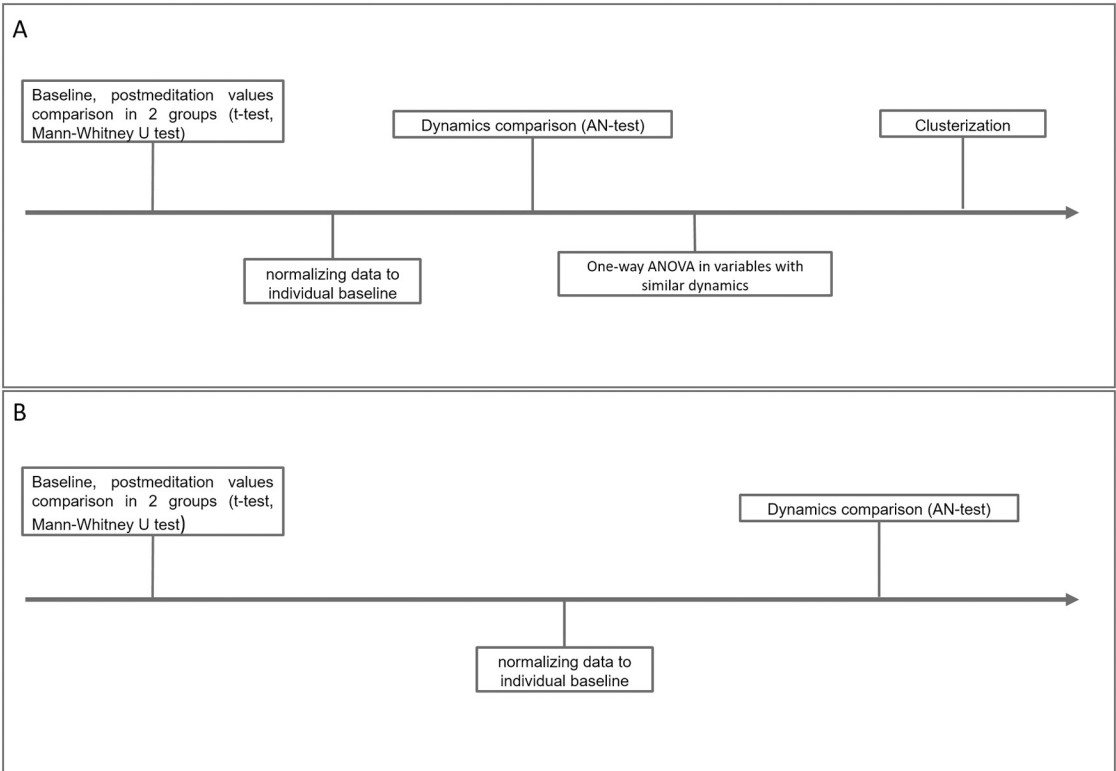

**Fig 2. Statistical analysis scheme.** (A) Comparison of the experienced meditators and the novices groups. (B) Comparison of two subgroups of the experienced meditators.

represent the data in a smaller dimensional space to increase the test power. Unlike linear regression, it also allows curves and the mean difference between the curves to be of arbitrary shape. The obtained *p*-values were then corrected for multiple comparisons using the FDR procedure [54].

The EEG indices that demonstrated different dynamics according to the AN-test were then compared separately at each stage between the three groups using one-way ANOVA test, followed by the post-hoc Tuckey's test. The obtained in the ANOVA test *p*-values were corrected for multiple comparisons using the B-H FDR procedure.

To test for between-stage differences, we applied one-way ANOVA followed by the post-hoc Tuckey's test to the indicators that had similar dynamics in the experienced meditators and novices, to describe common for both groups step-by-step changes during their meditation.

The index of relaxation (IR) was computed as the product of the coefficient of variation (CV) derived from PPG and the prefrontal alpha/theta ratio, i.e. IR = (pf_Alpha/Theta)*CV. This is a novel index that was not used previously. Forming it we wanted an index that would include a marker of mind relaxation/concentration and a marker of parasympathetic/sympathetic balance. The indices to include in the IR were originally chosen based on the literature review. CV is traditionally used to reflect the HRV associated with parasympathetic activity and peripheral relaxation [55]. The prefrontal alpha/theta ratio was used as a marker that was supposed to be decreased during meditation immersion. An increase of theta power was previously shown to be associated with a range of cognitive states and processes, such as regulation of focused attention [56], conscious awareness [57], meditation [58], sustained attention and

mental effort [59]. On the contrary, alpha rhythm desynchronization is usually found during active cognitive processes [60]. So, the decrease of alpha/theta ratio can be considered as an indicator of mind concentration and immersion [61]. The prefrontal location was chosen as one of the brain regions that was shown to be associated with meditation practice [62, 63].

It is also noteworthy that the indices IR comprises are practically appealing. HRV indices and prefrontal EEG can be easily measured now by wearable consumer devices, like fitness trackers and consumer level EEG headbands and therefore the IR index can be easily calculated.

In order to minimize between-subject variability in our IR index we use normalized with respect to the baseline values of the alpha/theta ratio and the CV index. According to the composition of the IR index K times decrease of mind relaxation index accompanied by K times increase of body relaxation (parasympathetic activity) index result in IR = 1. Increase or decrease of such index indicates a shift in mind concentration vs. body relaxation balance. We also would like to emphasize that using the IR index alone without exploring the values of its constituents may be misleading. For example, the IR index value equal to one can result both from no changes at all or simultaneous changes in the opposite directions of the two factors the IR comprises.

We compared the IR dynamics between the experienced meditators and the novices, using the adaptive Neyman test, see Supplementary materials. We also applied a clusterability test [64] to check for multimodality of the IR dynamical profiles. As described in the Results section, the clusterability test rejected the hypothesis of unimodality of the IR profiles in the experienced, but not novice meditators, and the corresponding two clusters were identified using the K-Means clustering, in the individual IR profiles of the experienced meditators. So, the experienced meditators group was divided into two subgroups. To perform statistical assessment of the dynamics of changes in the two subgroups we repeated our pipeline (Fig 2B) for each subgroup separately.

## Results

### Participants characteristics

We included 28 subjects in final analysis, i.e., 15 novices and 13 experienced meditators.

Two groups were similar in terms of age, body mass index (BMI), gender distribution, usual sleep duration, and sleep duration on the day of experiment. Median meditation experience in the experienced meditators group was 8 years of regular practice (from 2 to 32 years).

Questionnaire scores, blood pressure, and heart rate (HR) were similar both before and after the meditation. Also, the changes in all these indicators did not differ between groups (S1 Table).

ANS activity (S2 Table) and EEG indices did not differ significantly during the resting state baseline recorded prior to the meditation session between the novices and the experienced meditators. We used normalized to individual baseline values in the subsequent analysis.

### Intergroup comparison of step by step changes

We used the AN-test [53] to compare the dynamics of the individual indices. Importantly, all data were normalized to the individual baseline by dividing by premeditation resting state value. So, the resting state value was equal to one in all subjects. We found that the experienced meditators and the novices differed significantly in terms of changes of the most heart rate variability (HRV) derived indices, respiration rate, and respiration amplitude (S3 Table).

Meditators had consistently higher HRV indices (max RR, dRR, SDNN, CV), high frequency peak values, and lower stress index (SI) as compared to the novices within all meditation stages (detailed description of the HRV indices can be found in the Materials and methods section)

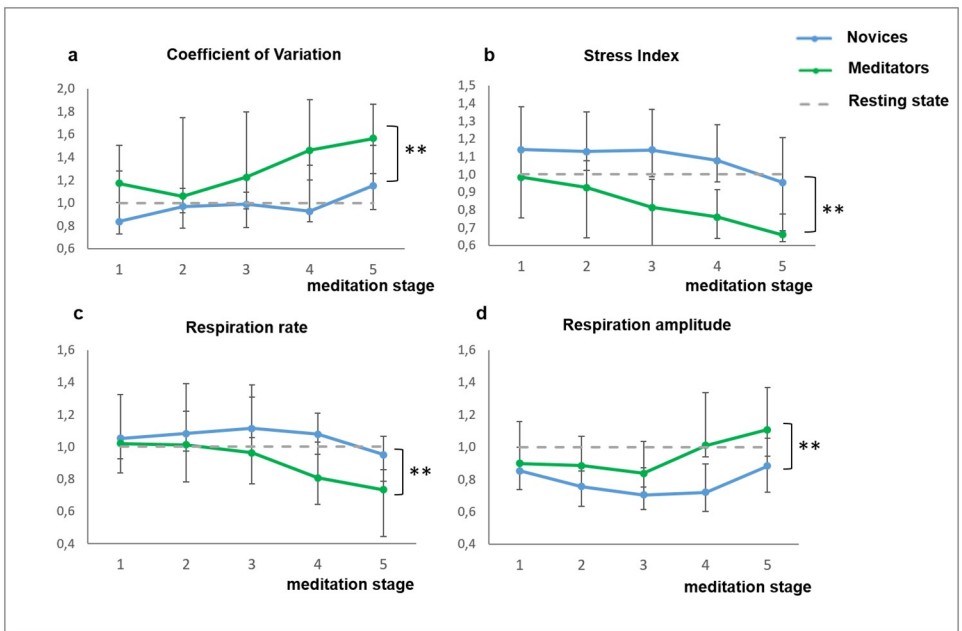

**Fig 3. Changes of ANS activity markers in the experienced meditators (n = 13) and novices (n = 15).** Data presented as Median ± IQR. **—significant intergroup difference according to AN-test (p<0.01). The dashed line reflects resting state reference level.

(Fig 3). Respiration rate was lower and the respiration amplitude was higher as compared to the novices starting from the third meditation stage (Fig 3).

Somewhat surprisingly, there was no significant difference in the averaged EEG-derived indices dynamics between the experienced meditators and the novices. One-way ANOVA analysis revealed the common changes that occurred during meditation in both experienced meditators and novice subjects. We have found a significant effect of the meditation stage factor for alpha power in several locations (left and right parietal and occipital) and beta power in most locations (S4 Table, Fig 4). In all cases EEG power decreased during the first half of meditation and bounced back to the premeditation level during the second part which is inline with our original hypothesis.

Overall, the dynamics of EEG indices in several leads follows a U-shaped curve taking its minimum at the second and third stages of meditation that included relaxation, body scan, and stopping internal dialogue instructions. This U-shape was more pronounced for beta power in most of the leads and alpha power in a relatively small group of leads.

Let us summarize the results obtained so far. Group level comparison revealed the increase of HRV (SDNN, CV) indices, decrease of respiration rate and increase of respiration amplitude in the experienced meditators group as compared to the novice. The latter effect seemed to be cumulative and became more pronounced by the last meditation stage. The EEG dynamics did not differ significantly between the two groups. There was a similar decrease of beta and alpha power in several leads during the meditation with the most pronounced effect in the midst of the meditation session.

## Clustering of data based on CV x prefrontal alpha/theta

So far, we obtained an evidence in support of our initial hypothesis about the relaxed alertness state in the experienced meditators, i.e., decrease of EEG power in alpha and beta band

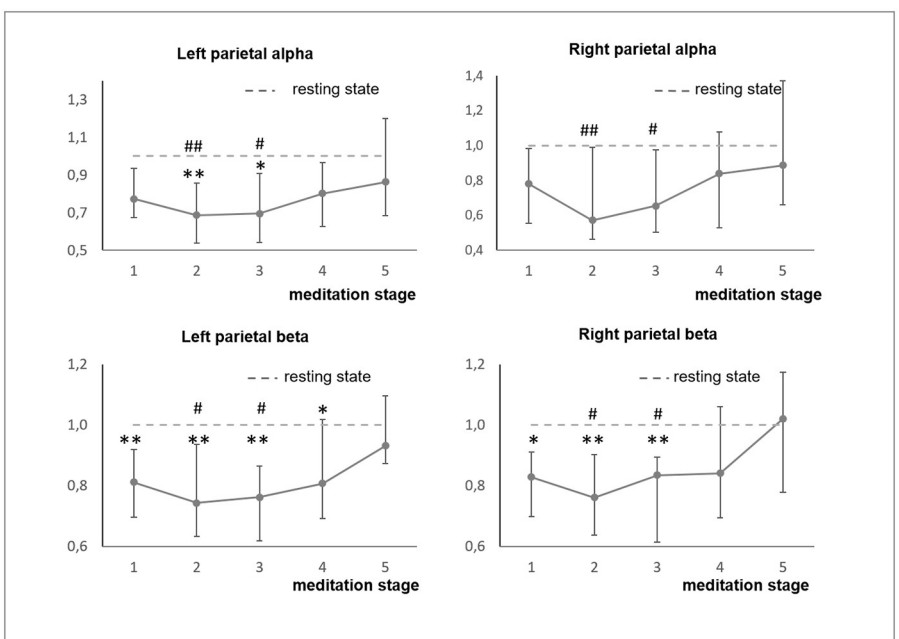

**Fig 4. Changes of EEG-derived indices in both groups together.** Changes of EEG-derived indices that had the same dynamics in the both groups according to the curve comparison test were analyzed using one-way ANOVA test. The obtained p-values were corrected for multiple comparisons using the FDR B-H procedure. Those that had a significant effect of "meditation stage" factor were analyzed by the post-hoc Tuckey test. Both groups combined (n = 28) were analyzed. Data presented as Median ± IQR. *, **—significant difference from resting state (p<0.05, p<0.01). #, ##-significant difference from the 5th meditation stage (p<0.05, p<0.01).

(compared to baseline, but not to the Novices), increase of HRV indices and decrease of respiration rate. Based on this data, we expected to see the simultaneous increase of mind concentration and peripheral relaxation markers common to the majority of the experienced meditators. To test this hypothesis, we introduced an index of relaxation (IR). As a scalar indicator of the relaxed alertness state, we used the product of the PPG derived coefficient of variation (CV) and the prefrontal alpha/theta ratio. The values of IR significantly below 1 corresponded to the reduced relaxation state, whereas the excessive relaxation corresponded to the values of IR>>1. An additional motivation behind this index can be found in "Methods" section.

The experienced meditators and novices exhibited significantly different IR dynamics (p<0.028, AN-test) (Fig 5a). Within the group of novice subjects, we have observed a unimodal dynamics of the IR. At the same time, the experienced meditators separated into two clearly defined subgroups (Fig 5b). Clusterability test revealed significant evidence (p = 0.016) to reject the hypothesis of unimodal distribution of the IR profiles in the experienced meditators and no such evidence (p = 0.781) in the novice group. Next, we applied K-Means clustering to the individual IR profiles of our subjects in an experienced group and assigned the profiles to the two clusters containing nearly equal numbers of subjects (7 and 6).

Based on the combination of the objective measurements reflecting the ANS and CNS activity, the first group of 7 subjects could be called "relaxed" meditators as their relaxation index increased during the meditation session. In contrast, the rest of the meditators (6 subjects), can be dubbed as the "concentrated" meditators since they are hallmarked with the decreasing IR profiles., The group of novice meditators had only one outlier and the rest of 14

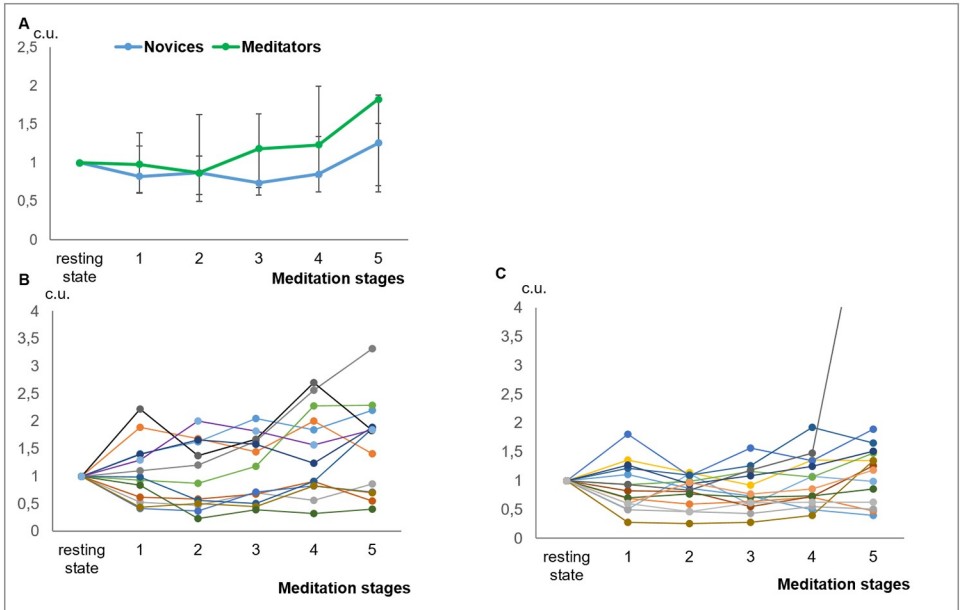

**Fig 5. The index of relaxation (CV\* prefrontal alpha to theta ratio) during meditation.** (A) in the Experienced meditators and Novices. AN-test revealed a significant difference of curves (p = 0.028). Data presented as Median ± IQR; (B), (C)–in individual Experienced and Novice subjects respectively.

subjects fell into a single cluster with the IR profiles grouped around 1 over the course of the meditation (Fig 5c).

## Comparison of the "relaxed" and "concentrated" meditators subgroups

We compared the "relaxed" and "concentrated" meditators groups. We first checked for and did not find statistically significant differences in terms of age (p = 0.67), gender (p = 0.13), meditation experience (p = 0.8), questionnaire scores (S5 Table), and habitual meditation style (Fig 6).

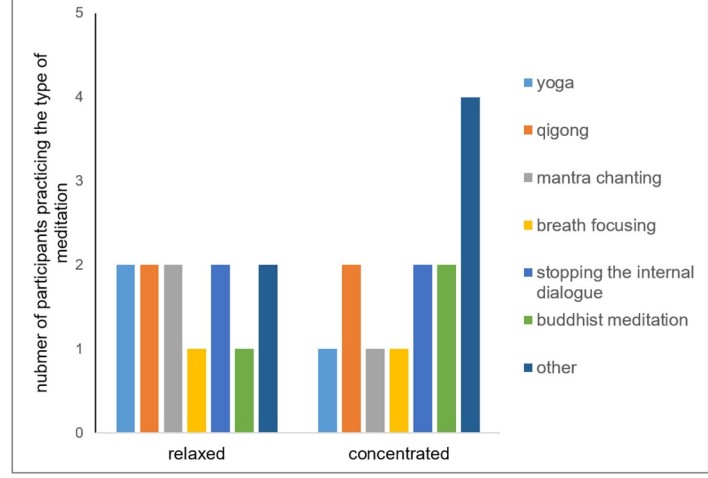

**Fig 6. Distribution of habitual meditation styles reported by participants in two subgroups.** Please note that most of the subjects reported more than one habitual meditation style.

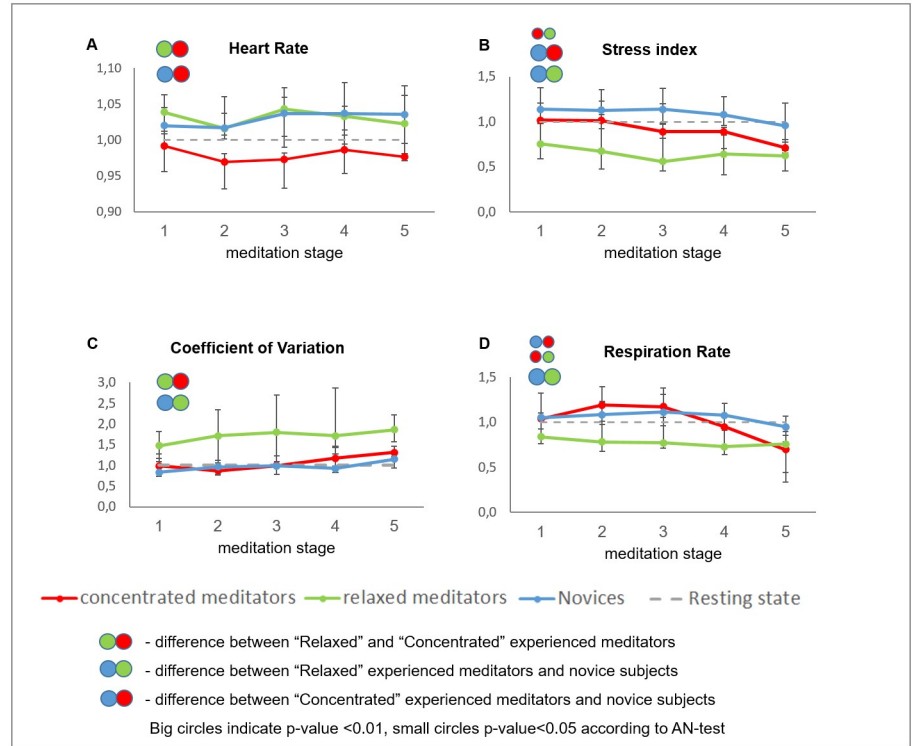

**Fig 7. Changes of ANS activity markers in "relaxed" and "concentrated" experienced meditators and novices.**
Data presented as Median ± IQR.

The subjects in these groups were also not different in terms of the EEG-derived indices and those reflecting the activity of the autonomic nervous system computed during the resting state baseline condition. Therefore, we can conclude that the observed changes were induced by the administered meditation session and the observed differences objectively reflected distinct strategies employed by the two subgroups.

Further analysis showed that these subgroups differed significantly during the meditation (S6–S8 Tables). The "relaxed" meditators demonstrated the increase of HRV family of indices as well as the decrease of the stress index and respiration rate (Fig 7), and also could be characterized by the decrease in the left frontal delta power with accompanying increase of the occipital and parietal alpha power and alpha/theta ratio (Figs 8 and 9).

In contrast, the "concentrated" meditators showed little changes in the HRV indices, the decrease of HR, and the increase of SI starting with the 3$^{rd}$ stage of meditation (Fig 7). In the EEG derived indices, we have observed the increase of delta power and strong decrease in alpha power and alpha/theta, and alpha/beta ratios during the meditation (Figs 8 and 9).

Detailed comparison of groups is given in the next section.

## Comparison of the relaxed and concentrated meditators subgroups with novices

We found the difference in the activity of ANS of the "relaxed" meditators vs. novices (S8 Table). In particular, in the relaxed meditators subgroup, we observed higher SDNN and CV, lower stress index and respiration rate, and the increased breathing amplitude as compared to the novices (Fig 7).

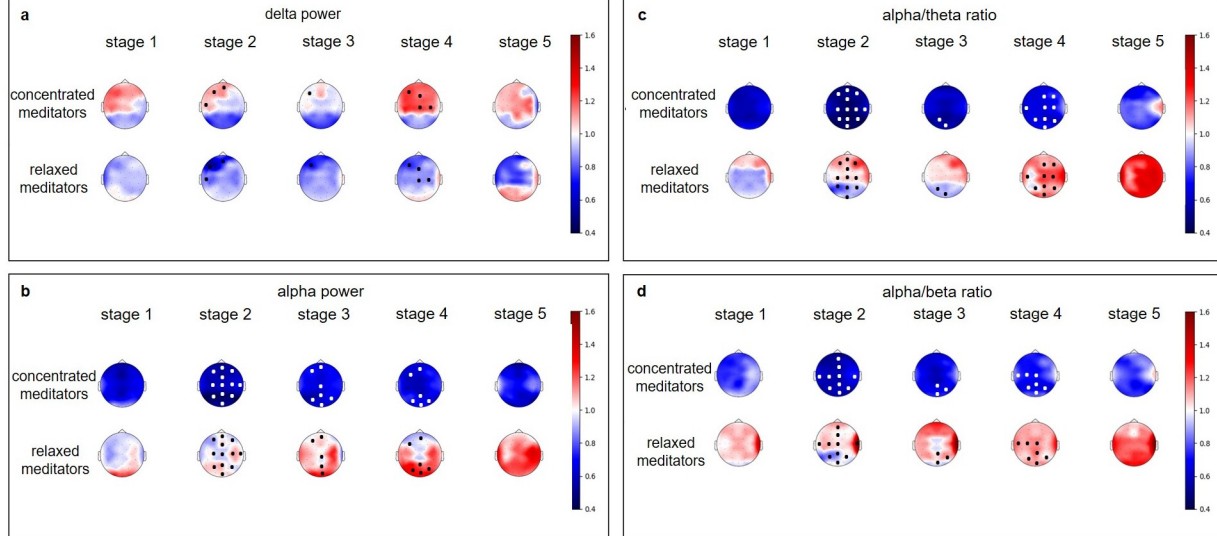

**Fig 8. Changes of EEG-derived indices in the "relaxed" and "concentrated" experienced meditators.** All values were normalized to the individual baseline. Data are presented as a group median value. We have used a common colorbar for all EEG indices and groups to make it easier to compare effects. Dots mark significant (p<0.05) difference between two groups according to the one-way ANOVA test, followed by the post hoc Tuckey's test. The colors of the dots were chosen to improve visibility and don't carry any additional information.

In the "relaxed" meditators group, the most pronounced significant decrease in delta power was found in the left frontal region. The index decreased gradually from the 1st to the 3rd meditation stage and increased during the following stages. Alpha power and alpha/theta ratio reached the maximum by the 5th stage. The most pronounced difference in alpha power and

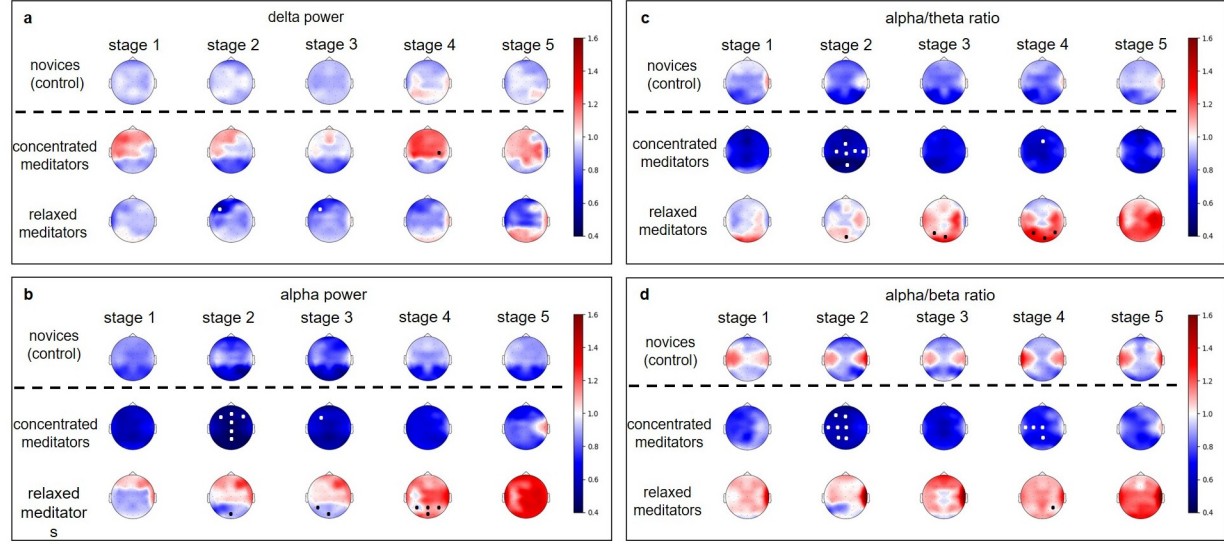

**Fig 9. Changes of EEG-derived indices in the "relaxed" and "concentrated" experienced meditators and novices.** All values were normalized to the individual baseline. Data are presented as a group median value. We have used a common colorbar for all EEG indices and groups to make it easier to compare effects. Dots mark significant (p<0.05) difference between Experienced meditators group and Novices according to the one-way ANOVA test, followed by the post hoc Tuckey's test. The colors of the dots were chosen to improve visibility and don't carry any additional information.

alpha/theta ratio was found in the left and right parietal and occipital regions (Fig 9, S6 and S7 Tables).

In general in the relaxed meditators, the overall modulation of EEG indices during the meditation session was weak.

Interestingly, in contrast, the "concentrated" meditators demonstrated fewer differences with novice subjects in the autonomic nervous system indices (SDNN and CV dynamics were similar in these two groups), but showed a dramatic decrease of alpha power, and alpha/theta and alpha/beta ratios in most of the electrode groups (Figs 7–9). Significant difference in the EEG derived indices dynamics between the novices and the concentrated meditators was found in the prefrontal, frontal, central, left and right frontal, left and right central, left temporal delta power, and alpha power in all regions with the exception of temporal and left parietal regions. Alpha to beta ratio dynamics differed in all brain regions, and alpha to theta in all but with the exception of right parietal and occipital. The "concentrated" meditators exhibited the increase in delta power and strong decrease in alpha power, and alpha/theta and alpha/beta ratios during meditation (Figs 8 and 9, S6 and S7 Tables).

Frontal, right and left frontal, left temporal delta increased in the "concentrated" meditators at the 1st stage of meditation, and then the increase of delta power repeated in the anterior (prefrontal, frontal, central) regions at the 4th stage of meditation. Delta activation was lateralized to the left.

For "concentrated" meditators, the strong decrease of alpha power was found in all brain regions. Alpha power decreased, reaching a minimum at the 2nd stage of meditation, and then increased gradually but still remained below the premeditation baseline at the 5th stage of meditation. The same holds for alpha/theta and alpha/beta ratios.

We discovered two distinct subgroups of the experienced meditators. Both groups differed from the novices, but in the first subgroup dubbed as "relaxed" experienced meditators, the difference was mostly due to peripheral indices and in the second, "concentrated" experienced meditators, mostly due to the EEG-derived indices. We summarized these observations in the following chart (Fig 10).

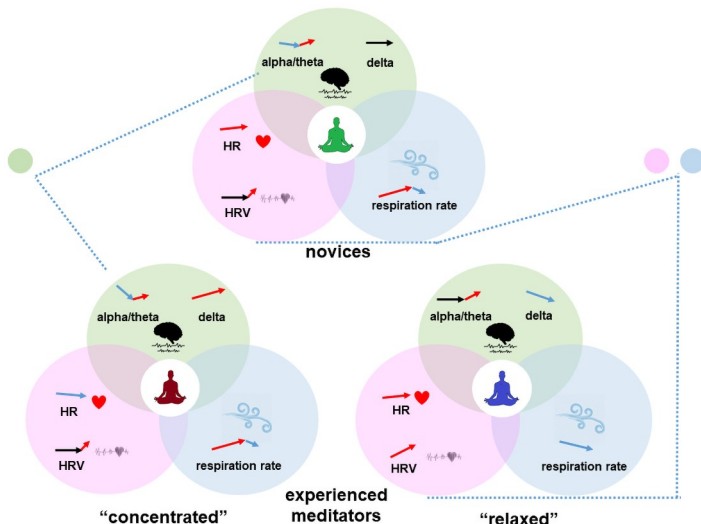

**Fig 10. Visual summary of the obtained results.** Arrows mark direction of changes of markers during meditation.

## Discussion

In this study, we used guided Taoist meditation administered with the stage-by-stage instructions. The CNS responses were assessed with EEG recordings and the ANS responses were assessed with polygraphic recordings including respirogram, photoplethysmography and the galvanic skin response. There were two experimental groups: the experienced meditators group–subjects with at least 2 years of regular meditation practice, and the novice group–subjects without experience of intentional meditation.

The main finding is the segregation of the experienced meditators into two subgroups, one of which had signs of relaxation during meditation and the other exhibited signs of concentration. Interestingly, no such effect was observed in the novices.

First, we compared ANS responses during meditation in the experienced meditators and novices groups. The groups differed significantly during the meditation in terms of several ANS markers. We found that the meditation effect on the heart rate variability (HRV) indices differed between the experienced and novice subjects. The HRV was higher and Baevsky Stress Index was lower in the experienced meditators than in the control group.

HRV analysis is one of the commonly used methods to assess the states of the sympathetic and parasympathetic nervous system. Primarily, time-domain indices such as SDNN, CV, RMSSD and the stress index differed between the groups in our study. Meditators demonstrated higher HRV indices and lower stress index as compared to the novices within all meditation stages. The increase of SDNN, CV and RMSSD in short-term recordings reflects primarily parasympathetic activation, while dRR (difference between highest and lowest RR intervals) is mostly sensitive to the changes in the respiration rate. It was shown previously that slower respiration rates (and longer exhalation) can produce higher respiratory sinus arrhythmia amplitudes, reflected by higher dRR, which is not dependent on vagal tone [65]. We also observed a decrease of the Stress Index reflecting activation of the central loop or sympathetic regulation in the experienced meditators [55].

Our data appeared to be consistent with the results of many previous studies, examining HRV during meditation and reporting the increase of parasympathetic activation during meditation. Previously, this effect was observed in novice subjects during meditation, compared to the non-meditating control group [36, 66] as well as in the experienced meditators compared to novices [67]. At the same time, several other studies found a decrease of vagal tone associated HRV indices during meditation [34, 49]. It was discussed previously that different types of meditation could change the balance of sympathetic and parasympathetic modulation in different ways. While some meditation types increase vagal tone and cause general relaxation, the others, on the contrary, lead to sympathetic system activation and general arousal [49, 68].

Our analysis of the respiration rate and amplitude also revealed significant differences between the experienced meditators and novices. The respiration rate decreased and respiration amplitude increased during the meditation in the experienced subjects. Similarly to the HRV index, the difference increased during the last stages. The slowing of respiration explains the increase of dRR index as a slow respiration rate normally produces a high level of respiratory sinus arrhythmia [65]. Our observation here is in agreement with the previously reported attenuation of respiration rate, commonly observed during meditation, except for the techniques where explicit opposite instructions are given [41, 42].

Next, we explored EEG during meditation. The detailed review by Lomas and colleagues [30] summarized that mindfulness was most commonly associated with the enhanced alpha and theta power. Such simultaneous increase of alpha and theta power can indicate a "relaxed alertness" state or "tonic alertness" [35]. Data about beta activity are even more inconsistent.

Beta band power appeared unchanged during meditation in one study [69], while decreased [47, 70, 71] or increased in others [72, 73].

It was found that the attenuation of beta activity correlated with the elevation of the BOLD signal in precentral motor areas [74] and with facilitating sensory processing of attended somatosensory stimuli [75]. Therefore, the decreased beta power can be interpreted as a sign of enhanced cortical activation [35]. The decrease of alpha activity can indicate attenuation in mind wandering [76] which is one of the main goals of meditation.

Our direct comparison of EEG changes during meditation in the experienced meditators and novice subjects did not reveal any significant difference between the two groups. One-way ANOVA analysis of the pulled data of both groups revealed U-shape changes in spatially distributed beta power indices and also in alpha power profiles over the left and right parietal and occipital regions. These indices decreased during the first half of meditation and returned to the baseline during the second half. Potentially, this can be explained by the fact that both the experienced meditators and novice subjects were able to enter into the "presence in the moment" state [70] as dictated by the audio instruction.

Our initial comparison of dynamics of the ANS activity measures and the EEG indices revealed the decrease of alpha and beta power in the experienced meditators and novice subjects, which can indicate decreased mind wandering and presence of the self-awareness state, focused on the present experience. In the experienced meditators, these changes were accompanied by the signs of elevated parasympathetic modulation and peripheral relaxation. These findings are in line with our preliminary assumption about the "relaxed alertness" state present in the experienced meditators. To check it further, we introduced the index of relaxation (IR) that we calculated as a scalar product of PPG-derived coefficient of variation (as a commonly used marker of parasympathetic activity and peripheral relaxation) and prefrontal alpha/theta ratio (as a marker of immersion/concentration [61]. Prefrontal alpha/theta was chosen based on previous data about the strong impact of meditation on the prefrontal cortex [62, 77].

Contrary to our expectations, we found no difference in the average index of relaxation (IR) between the two groups. We then hypothesized that the lack of such a difference between the experienced and novice meditators comes from a large within group variation. The clusterability test applied to the IR profiles revealed significant multimodality in the IR profiles in the experienced meditators group. The subsequent visual analysis and K-means clustering of the IR profiles reveals the obvious split of the experienced meditators group into two separate subgroups. In one of them, the IR increased significantly during the meditation, while in the other, the IR did not change much during the entire meditation session. Intriguingly, no such split could be seen in the novices. For convenience, we called the subgroup of the experienced meditators, who demonstrated the increased IR,–"relaxed" meditators and the other subgroup "concentrated" meditators.

Interestingly, the two subgroups of the experienced meditators did not differ in terms of age, gender distribution, duration of regular meditation practice or habitual meditation style. In addition, the physiological indicators we measured did not differ between the "relaxed" and the "concentrated" experienced meditators in the resting state.

However, we observed very pronounced diverging dynamics of the physiological indicators during the actual meditation. "Concentrated" meditators demonstrated an increase in delta power, a pronounced decrease of alpha power, and alpha/beta and alpha/theta ratios and a slight decrease in respiration rate at the last stages of meditation and the decrease of HR. The "relaxed" meditators, on the contrary, had a strong increase of all HRV indices, surprising elevation of the HR, a decrease of respiration rate and an increase of respiration amplitude, an increase or no changes of alpha and beta power and alpha/theta and alpha/beta ratios.

As shown previously, the increased frontal delta power can be associated with attentional engagement [78] and internal attention processes [79]. There are two opposite hypotheses about the role of alpha rhythm in cognitive processing. The first one states that the increased alpha oscillations reflect inactivity of the underlying neural substrate, the second one states that alpha oscillations play a big role in the performance of internally directed tasks, by inhibiting excessive internal information [80]. At the same time, increased alpha power was associated with mind wandering episodes [76, 81]. Alpha oscillations suppression was related to somatosensory stimulus awareness [82]. Our data biases us towards interpreting alpha power decrease as a sign of internal attentional processes.

Thus, we mostly observed relaxation, as judged by both CNS and ANS activity indicators, in the half of the experienced meditators, but a concentration effect without any additional tension in the second half of the experienced meditators group.

Our findings contribute to the fundamental discussion about the general effects of meditation and that it may result either into relaxation or arousal [35]. As was discussed previously [34], different types of meditation can have different effects. For the first time, we revealed the presence of both strategies of meditation among the experienced meditators despite the fact that all the subjects performed the same instruction. At the same time, both subgroups of the experienced meditators differed from the novice subjects. Thus, we clearly observed an objective evidence of some special skill of submersion into meditation, possessed by the experienced meditators, that is probably dependent on their previous experience. As most of the subjects reported a regular practice of several types of meditation, we cannot make any conclusion about the association between the habitual style of meditation and the strategy. We also have no reasons to make a judgement regarding the quality of the two strategies since the subjects comprising both groups, on average, had the same and significant duration of meditative practice. Therefore, we can assume that people that implement regular meditative practice in their everyday life could choose one of the two strategies of meditation, relaxation or arousal, and follow the selected path, even when following identical instructions. Whether or not this choice is conscious and could be controlled is also unclear. This observed dichotomy in the group of the experienced meditators may partly explain the diversity and incongruence of the earlier reports on the observed changes in physiological indicators during the meditation.

Our results are also valuable in the context of development of the high-end biofeedback systems for assisted meditation. Our observation that physiological changes that happen during meditation do not exclusively depend on meditation guidelines but are also related to some individual predispositions renders the use of the common biofeedback algorithm useless or even potentially harmful. Preliminary identification of the individual meditation strategy of a particular subject is necessary to tailor subject-specific guidelines. Additionally, even if the same guidelines are used after the meditation session, using the IR index introduced here, the subject could be informed regarding the strategy he consciously, or not so consciously, followed. Both ANS and CNS parameters are necessary to record in order to tailor the effective biofeedback strategies, in accordance with individual predisposition. Moreover, ANS and CNS coupling could serve as an additional metric to be optimized during a feedback session.

## Conclusions

During guided meditation, experienced meditators exhibited physiological changes that were different from those found in novice subjects. Two meditation strategies were identified in the experienced meditators using a cluster analysis. While one subgroup of long-term meditators exhibited changes indicative of relaxation, such as increased HRV indices, decreased respiration rate and increased alpha power and alpha/beta and alpha/theta ratio, the other subgroup,

conversely, had signs of mind concentration and immersion without peripheral relaxation. The meditation strategy did not depend on the previous meditation experience, age of subject or other measured indicators. There was no clustering into subgroups in the novice subjects. The study also confirmed the previous reports of an increase in CNS and ANS interaction in experienced meditators.

## Supporting information

**S1 Table. Questionnaire data.**
(PDF)

**S2 Table. ANS activity markers during the resting state.**
(PDF)

**S3 Table. Comparison of ANS activity markers changes in the experienced meditators and the novices.**
(PDF)

**S4 Table. Changes of EEG-derived indices that had same dynamics in the both groups.**
(PDF)

**S5 Table. Questionnaire data in "concentrated" and "relaxed" experienced meditators.**
(PDF)

**S6 Table. Comparison of EEG-derived indices changes in the "relaxed" and "concentrated" experienced meditators and the novices, according to the AN test.**
(PDF)

**S7 Table. Comparison of EEG-derived indices changes in the "relaxed" and "concentrated" experienced meditators and the novices, according to the one-way ANOVA test.** The EEG indices that demonstrated different dynamics according to the AN-test were then compared separately on the each stage between the three groups using one-way ANOVA test, followed by the post-hoc Tuckey's test. The obtained in the ANOVA test p-values were corrected for multiple comparisons using the FDR B-H procedure.
(XLSX)

**S8 Table. Comparison of ANS activity markers changes in the "relaxed" and "concentrated" experienced meditators and the novices, according to AN test.**
(PDF)

**S9 Table. Questionnaire.** Individual data.
(XLSX)

**S10 Table. Individual EEG data.** Data presented as EEG power and EEG power ratios for every subject. EEG power spectra were computed using the Welch method, using a window size of 1s with no overlap for each meditation stage separately. Next, we averaged power spectral density profiles within the following five bands: Delta: 1–4 Hz, Theta: 4–8 Hz, Alpha: 8–14 Hz, Beta: 14–25 Hz, and Gamma: 25–40 Hz. Additionally, using the obtained stage specific mean band power values, we also computed alpha/theta and alpha/beta EEG power ratios.
(XLSX)

**S11 Table. Individual respiration, PPG and GSR data.** Data presented as ANS activity indices for every subject. Full description of the calculated indices can be found in the 'Experimental Design, Materials, and Methods' section.
(XLSX)

## Acknowledgments

We thank Olga Tyurina and Aleksey Aleksentsev for assistance with the meditation guideline adaptation and recording.

## Author Contributions

**Conceptualization:** Maria Volodina, Alexei Ossadtchi.

**Data curation:** Maria Volodina, Alexei Ossadtchi.

**Formal analysis:** Nikolai Smetanin, Alexei Ossadtchi.

**Funding acquisition:** Mikhail Lebedev, Alexei Ossadtchi.

**Methodology:** Maria Volodina, Nikolai Smetanin, Alexei Ossadtchi.

**Software:** Nikolai Smetanin.

**Supervision:** Alexei Ossadtchi.

**Validation:** Alexei Ossadtchi.

**Visualization:** Alexei Ossadtchi.

**Writing – original draft:** Maria Volodina.

**Writing – review & editing:** Mikhail Lebedev, Alexei Ossadtchi.

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
