## [Decision Letter · Decision Letter 0]

20 Aug 2021

Bélatelep, Hungary

July 17, 2021

PONE-D-21-17960

Cortical and autonomic responses during staged Taoist meditation: two distinct meditation strategies

PLOS ONE

Dear Dr. Ossadtchi,

Thank you for submitting your manuscript to PLOS ONE. After careful consideration, we feel that it has merit but does not fully meet PLOS ONE’s publication criteria as it currently stands. Therefore, we invite you to submit a revised version of the manuscript that addresses the points raised by the Reviewer, listed below.

We look forward to receiving your revised manuscript.

Kind regards,

Joseph Najbauer, Ph.D.

Academic Editor

PLOS ONE

Journal Requirements:

Reviewers' comments:

Reviewer's Responses to Questions

**Comments to the Author**

1. Is the manuscript technically sound, and do the data support the conclusions?

Reviewer #1: Yes

2. Has the statistical analysis been performed appropriately and rigorously? 

Reviewer #1: No

3. Have the authors made all data underlying the findings in their manuscript fully available?

Reviewer #1: No

4. Is the manuscript presented in an intelligible fashion and written in standard English?

Reviewer #1: Yes

5. Review Comments to the Author

Reviewer #1: For several reasons this is an interesting paper concerning the physiological correlates of meditation. It is rare that a form of guided Taoist meditation is investigated with a set of physiological indices. Two subgroups of meditators are identified through their respective physiological markers. I have a few remarks/suggestions which could be taken up (in a discussion, as limitations), in order of appearance in the text:

What is a little odd (unseen so far) is that experienced meditators of various backgrounds (but only 1 Taoist meditator) meditate in a practice not experienced in. Although the authors argue that this type of meditation falls within the category of mindfulness meditation, there are quite a few differences. The odd thing is that experienced meditators do a meditation practice they are new to. This fact, the underlying rationale, should be taken up in the introduction.

Lines 75ff: The categorization between mindfulness (which is a secular construct) and transcendental (which is a special spiritual practise) is confusing. The later distinction between focused and open monitoring is more to the point.

Lines 88ff: At the end of the sentence (line 90) one would have expected references.

Lines 123ff: The hypothesis (“we expected”) is confusing. At least it must be explained or backed up by previous studies. Why would the physiological markers gradually return to normal after the middle of the meditation? This looks more like a hypothesis “after the fact”. Related, elaborate on this finding in the results section and moreover in the discussion. Is it that meditators were not experienced enough or that the novel type of meditation was disturbing? An experienced meditator could hold an effect for 30 minutes or peak at the time. By the way, make explicit that the meditation session lasted for 29 or 30 minutes.

Stage 2 is an unusual instruction. It is more suited for medical students who meditate (does everyone know where the sacrum or the cervical vertebra are?)… Nobody could accurately stretch the 12th, 11th, etc., vertebra. Explain. This is done as an approximate way and functioning as a counting strategy?

Stage 11: what do you do when you don’t know what qi or shen means? This relates to a limitation section where one has to discuss whether and how people could commit themselves to this special type of meditation.

Lines 351f: Are there more indices for meditation experience. Often one uses “meditates x hours a week for the last 8 weeks”. Otherwise, someone meditating once a day for three hours and someone meditating once a month for 20 minutes would both have a regular practice.

Lines 402: you did not find support for a decrease in EEG power in experienced meditators.

Lines 405ff: The IR index is reported superficially. Why these indices (PPG, alpha-theta ratio) and why the product? Has this been done before?

Fig. 5, C: there is one outlier. Is this a measurement artefact?

Page 23, Fig. 7, and text: physiologically speaking the “relaxed meditators” seem like the successful meditators while the “concentrated meditators” are like the novices.

Page 26: All the reported differences, are they significant? Just showing decreases and increases and colour differences in Fig. 8 is not enough.

6. PLOS authors have the option to publish the peer review history of their article (what does this mean?). If published, this will include your full peer review and any attached files.

Reviewer #1: No

---

## [Author Response · Author response to Decision Letter 0]

29 Sep 2021

What is a little odd (unseen so far) is that experienced meditators of various backgrounds (but only 1 Taoist meditator) meditate in a practice not experienced in. Although the authors argue that this type of meditation falls within the category of mindfulness meditation, there are quite a few differences. The odd thing is that experienced meditators do a meditation practice they are new to. This fact, the underlying rationale, should be taken up in the introduction.

Thank you very much for the important suggestion! We used such an unusual design intentionally. One of the questions we aimed to answer was if people who practice meditation regularly develop some special skill of controlling their inner state. We totally agree that in the vast majority of published studies subjects perform habitual practice during the experiment. Such a design, however, does not allow us to say anything about their ability to consciously control the physiological state in any situation other than this particular meditation practice. In such a standard design there is also some bias caused by the fact that the experienced meditators are likely to be familiar with the text of instructions while the subjects from the control group are most probably not. 

We placed the corresponding clarification to the introduction section, please see lines 120-124.

Lines 75ff: The categorization between mindfulness (which is a secular construct) and transcendental (which is a special spiritual practise) is confusing. The later distinction between focused and open monitoring is more to the point. 

Thank you! Such a categorization is indeed not so common but nevertheless was used in a previous study (Travis, F., & Shear, J. (2010, December). Focused attention, open monitoring and automatic self-transcending: Categories to organize meditations from Vedic, Buddhist and Chinese traditions. Consciousness and Cognition. Conscious Cogn. https://doi.org/10.1016/j.concog.2010.01.007). However, we agree that this can be confusing and we have removed this statement from the text, please see lines 74-79.

Lines 88ff: At the end of the sentence (line 90) one would have expected references. 

Thanks for spotting this! We now have placed the reference to the appropriate review by Lomas, T., Ivtzan, I., & Fu, C. H. Y. (2015). A systematic review of the neurophysiology of mindfulness on EEG oscillations. Neuroscience and Biobehavioral Reviews, 57, 401–410. https://doi.org/10.1016/j.neubiorev.2015.09.018 for review, please see line 90. 

Lines 123ff: The hypothesis (“we expected”) is confusing. At least it must be explained or backed up by previous studies. Why would the physiological markers gradually return to normal after the middle of the meditation? This looks more like a hypothesis “after the fact”. Related, elaborate on this finding in the results section and moreover in the discussion. Is it that meditators were not experienced enough or that the novel type of meditation was disturbing? An experienced meditator could hold an effect for 30 minutes or peak at the time. By the way, make explicit that the meditation session lasted for 29 or 30 minutes. 

Thanks for the request for clarification. Although we did not formally test the U-shape hypothesis we have generated it prior to collecting any data. This hypothesis was partly based on the subjective reports of experienced meditators and meditation teachers about gradual entering into meditative state and gradual emerging. Standard recommendation by meditation teachers is similar duration of entering and emerging. Also, the U-shape profile hypothesis appears to be consistent with the data by DeLosAngeles et al., 2016 who obtained such U-shape changes of theta, alpha and beta power in their study (doi: 10.1016/j.ijpsycho.2016.09.020. Epub 2016 Oct 1. PMID: 27702643.) At the same time we agree that such an expectation is not a common place, especially given a relatively small number of studies reporting staged meditation. Per your suggestion and to better serve our readers we included an additional justification of this hypothesis, please see lines 131-136. 

Stage 2 is an unusual instruction. It is more suited for medical students who meditate (does everyone know where the sacrum or the cervical vertebra are?)… Nobody could accurately stretch the 12th, 11th, etc., vertebra. Explain. This is done as an approximate way and functioning as a counting strategy? 

We totally agree that nobody, including the experienced meditators (at least most of them), can’t feel a particular vertebra. This instruction is designed to help the subject gradually move attention along the spine, stretching it up. Using sequential naming of spine segments helps to control the pace at which the subjects move their attention.

We have foreseen the possible problem with understanding the anatomical terms. Therefore we have tested the text in pilot experiments and found out the most common difficulties. To avoid the problem all the participants have read the text of meditation before the experiment and could ask any questions they have. We have also clarified the most common questions with all participants, like “Do you know where the sacrum is”, “Do you know that vertebrae are numbered from top to bottom?” and so on.

We added the corresponding clarification to the revised version of the manuscript, please, see lines 226-233. 

Stage 11: what do you do when you don’t know what qi or shen means? This relates to a limitation section where one has to discuss whether and how people could commit themselves to this special type of meditation. 

Thank you! As we mentioned above we discussed with all participants the text of meditation and unknown terms before the experiment. We have also briefly explained the meaning of ‘qi’ and ‘shen’ according to the recommendations of еру meditation instructor. But definitely most of the novices didn’t have a deep understanding of these terms that could be confusing. We now discuss this point in the “Methods” section, please see lines 226-233.

Lines 351f: Are there more indices for meditation experience. Often one uses “meditates x hours a week for the last 8 weeks”. Otherwise, someone meditating once a day for three hours and someone meditating once a month for 20 minutes would both have a regular practice. 

We have asked participants about the time they spend practicing. We didn’t include this information in the supplementary data initially as some of the participants could answer very roughly, e.g. “10-60 min daily” or “3-4 hours 2-3 times per week”. We calculated approximate time of practice subjects spent weekly (e.g. 35 min daily = 245 min weekly in the first case or 3.5 hrs 2.5 times per week = 525 minutes weekly in the second). We compared two groups (“relaxed” and “concentrated” meditators) using such estimation and didn’t find a significant difference (p = 0.95, t-test). We didn’t discuss this data in the text but we added the information on the estimated duration of practice per week to the supporting information, table S5. 

Lines 402: you did not find support for a decrease in EEG power in experienced meditators. 

Thanks! Indeed our statement about EEG power decrease was confusing. We obtained decrease of EEG power both in the experienced meditators and novices compared to baseline and this trend was similar in the experienced meditators and novices. As we mention in the text, at this stage of analysis (before the meditation groups splits in two) it looked like these EEG changes are characteristic of everyone who follows the instructions, regardless of their previous meditation experience. At the same time the accompanying changes of ANS activity did differ between groups. To ameliorate this lack of clarity we modified the corresponding text, please see line 437-438.

Lines 405ff: The IR index is reported superficially. Why these indices (PPG, alpha-theta ratio) and why the product? Has this been done before? 

Thank you! Indeed the IR index was not used so far therefore we do agree that its use merits some additional description of the motivation behind it. 

We aimed to introduce an index that would include a marker of mind relaxation/concentration and marker of parasympathetic/sympathetic balance. The indices to include in IR were chosen firstly based on literature data about changes in prefrontal brain areas (Gotink et al., 2016) caused by meditation. Alpha/theta ratio was chosen as the index of mind relaxation (decrease of concentration) (Lim et al., 2019) and CV index as a traditionally used marker of parasympathetic ANS activity (Baevsky, R. М. & Chernikova, 2017). 

In order to minimize between-subject variability in our IR index we use normalized with respect to the baseline values of the alpha/theta ratio and the CV index. According to the composition of the IR index K times decrease of mind relaxation index accompanied by K times increase of body relaxation (parasympathetic activity) index result in IR = 1. Increase or decrease of such index indicates a shift in mind concentration vs. body relaxation balance. We also would like to emphasize that using the IR index alone without exploring the values of its constituents may be misleading. For example, the IR index value equal to one can result both from no changes at all or simultaneous changes in the opposite directions of the two factors the IR comprises.

It is also noteworthy that the indices IR comprises are practically appealing. HRV indices and prefrontal EEG can be easily measured now by wearable consumer devices, like fitness trackers and consumer level EEG headbands and therefore the IR index can be easily calculated. 

We placed more detailed motivation in the “Methods” section and also included the corresponding statement in the “Results” section, please see lines 348-372 and 444-445.

Fig. 5, C: there is one outlier. Is this a measurement artefact?

Thank you very much for spotting it! Such an outlying value in the subject is due to the high alpha power in the last stage. There are no visible artefacts on the recording or comments in the lab journal that can explain such value. Thus, we can only assume some individual reaction to the meditation practice that led to such an increase in alpha power. This however does not seem to affect the main conclusion of the manuscript. 

Page 23, Fig. 7, and text: physiologically speaking the “relaxed meditators” seem like the successful meditators while the “concentrated meditators” are like the novices.

This is true for ANS activity indices (excepting heart rate), but not for EEG indices (Fig.8). Therefore as we discuss it is hard to actually decide which subgroup of the experienced meditators is performing more properly, please see lines 657-659. 

Page 26: All the reported differences, are they significant? Just showing decreases and increases and colour differences in Fig. 8 is not enough. 

Thank you for your comment. We changed the figures and marked significant differences between groups according to one-way ANOVA test, followed by post-hoc Tuckey’s test, see Figures 8-9. We also added Table S7 with p-values, according to this test and included description of analysis in ‘Methods’ section, please see lines 339-342.

The differences between all three groups were significant for most of the areas. Thus it looks very messy to mark significant areas on the figure. Full list of the p-values can be found in the supplementary materials (Table S6). We added the clarification and the links to the supplementary materials in the text, please see lines 478, 481-482, 486, 492-497, 499-505, 508-527.

---

## [Decision Letter · Decision Letter 1]

15 Nov 2021

Pécs, Hungary

November 12, 2021

Cortical and autonomic responses during staged Taoist meditation: two distinct meditation strategies

PONE-D-21-17960R1

Dear Dr. Ossadtchi,

We’re pleased to inform you that your manuscript (R1 version) has been judged scientifically suitable for publication and will be formally accepted for publication once it meets all outstanding technical requirements.

Kind regards,

Joseph Najbauer, Ph.D.

Academic Editor

PLOS ONE

Reviewers' comments:

Reviewer's Responses to Questions

**Comments to the Author**

1. If the authors have adequately addressed your comments raised in a previous round of review and you feel that this manuscript is now acceptable for publication, you may indicate that here to bypass the “Comments to the Author” section, enter your conflict of interest statement in the “Confidential to Editor” section, and submit your "Accept" recommendation.

Reviewer #1: All comments have been addressed

2. Is the manuscript technically sound, and do the data support the conclusions?

Reviewer #1: Yes

3. Has the statistical analysis been performed appropriately and rigorously? 

Reviewer #1: Yes

4. Have the authors made all data underlying the findings in their manuscript fully available?

Reviewer #1: Yes

5. Is the manuscript presented in an intelligible fashion and written in standard English?

Reviewer #1: Yes

6. Review Comments to the Author

Reviewer #1: I am fine with the responses to my earlier remarks. This study is somewhat novel, i.e. regrding the type of meditation and the results which point to two different strategies of meditation technique.

7. PLOS authors have the option to publish the peer review history of their article (what does this mean?). If published, this will include your full peer review and any attached files.

Reviewer #1: No

---

## [Editor Report · Acceptance letter]

22 Nov 2021

PONE-D-21-17960R1 

Cortical and autonomic responses during staged Taoist meditation: two distinct meditation strategies 

Dear Dr. Ossadtchi:

I'm pleased to inform you that your manuscript has been deemed suitable for publication in PLOS ONE. Congratulations! Your manuscript is now with our production department. 

Kind regards, 

on behalf of

Dr. Joseph Najbauer 

Academic Editor

PLOS ONE